# Patient experience of NHS health checks: a systematic review and qualitative synthesis

Juliet A Usher-Smith,[1] Emma Harte,[2] Calum MacLure,[2] Adam Martin,[3] Catherine L Saunders,[1] Catherine Meads,[4] Fiona M Walter,[1] Simon J Griffin,[1,5] Jonathan Mant[1]

► Prepublication history and additional material are available. To view these files, please visit the journal online (http://dx.doi.org/10.1136/bmjopen-2017-017169).

[1]The Primary Care Unit, Institute of Public Health, University of Cambridge, Cambridge, UK
[2]RAND Europe, Westbrook Centre, Cambridge, UK
[3]Academic Unit of Health Economics, Leeds Institute of Health Sciences, University of Leeds, Leeds, UK
[4]Faculty of Health, Social Care and Education, Anglia Ruskin University, Cambridge, UK
[5]MRC Epidemiology Unit, University of Cambridge, Institute of Metabolic Science, Cambridge, UK

**Correspondence to**
Dr Juliet A Usher-Smith;
jau20@medschl.cam.ac.uk

## ABSTRACT

**Objective** To review the experiences of patients attending NHS Health Checks in England.

**Design** A systematic review of quantitative and qualitative studies with a thematic synthesis of qualitative studies.

**Data sources** An electronic literature search of Medline, Embase, Health Management Information Consortium, Cumulative Index of Nursing and Allied Health Literature, Global Health, PsycInfo, Web of Science, OpenGrey, the Cochrane Library, National Health Service (NHS) Evidence, Google Scholar, Google, Clinical Trials.gov and the ISRCTN registry to 09/11/16 with no language restriction and manual screening of reference lists of all included papers.

**Inclusion criteria** Primary research reporting experiences of patients who have attended NHS Health Checks.

**Results** 20 studies met the inclusion criteria, 9 reporting quantitative data and 15 qualitative data. There were consistently high levels of reported satisfaction in surveys, with over 80% feeling that they had benefited from an NHS Health Check. Data from qualitative studies showed that the NHS Health Check had been perceived to act as a wake-up call for many who reported having gone on to make substantial lifestyle changes which they attributed to the NHS Health Check. However, some had been left with a feeling of unmet expectations, were confused about or unable to remember their risk scores, found the lifestyle advice too simplistic and non-personalised or were confused about follow-up.

**Conclusions** While participants were generally very supportive of the NHS Health Check programme and examples of behaviour change were reported, there are a number of areas where improvements could be made. These include greater clarity around the aims of the programme within the promotional material, more proactive support for lifestyle change and greater appreciation of the challenges of communicating risk and the limitations of relying on the risk score alone as a trigger for facilitating behaviour change.

## INTRODUCTION

The National Health Service (NHS) Health Check programme is one of the largest current prevention initiatives in England. Introduced in 2009 to improve cardiovascular disease (CVD) risk factors through behavioural

### Strengths and limitations of this study

► This is the first study to systematically review quantitative and qualitative studies that consider the experiences of patients who have attended NHS Health Checks.
► The use of broad inclusion criteria and the systematic search of multiple databases and the grey literature allowed us to include studies that had not been published in peer-reviewed scientific journals.
► The included studies were of varying quality.
► The quantitative studies reporting responses to surveys had response rates between 23% and 43%, making them at risk of responder bias.
► The qualitative studies included small, selected groups of participants whose expressed views were likely to be affected by both recall bias and social desirability bias.

change and treatment informed by risk stratification, it became a mandated public health service in 2013. Local authorities are now responsible for offering an NHS Health Check to individuals aged 40–74 without existing cardiovascular disease, diabetes or hypertension every 5 years. The NHS Health Check itself consists of three components: risk assessment, communication of risk, and risk management.[1] For CVD the QRISK2 risk tool[2] is first used to estimate the individual's risk of developing CVD based on risk factors including age, sex, ethnicity, smoking status, height and weight, family history of coronary heart disease, blood pressure and cholesterol. That estimated risk, expressed as the percentage risk of developing disease over the next 10 years, is then used to raise awareness of relevant risk factors and inform discussion about the lifestyle and medical approaches best suited to managing the individual's risk of disease. Risk assessment for diabetes was introduced in 2016 and patients at high risk of developing type 2 diabetes who should receive a screening blood test are identified by

using either validated risk assessment tools or a diabetes filter.[1] Based on modelling studies of cross-sectional data, it was estimated that the programme could prevent 1600 heart attacks and strokes, at least 650 premature deaths and over 4000 new cases of diabetes each year with an estimated cost per quality adjusted life year of approximately £3000[3]. However, whether NHS Health Checks represent an efficient use of scarce health promotion resources has been questioned.[4 5]

Alongside clinical effectiveness and safety, patient experience is increasingly recognised worldwide as one of the three elements of high-quality healthcare.[6–8] As well as enabling a better understanding of current problems with healthcare delivery, informing continuous improvement and redesign of services and helping professionals reflect on practice, a recent systematic review has shown that patient experience is positively associated with: self-rated and objectively measured health outcomes; adherence to recommended medication and treatments; preventative care; healthcare resource use; technical quality-of-care delivery; and adverse events.[9 10] There is also an association at the organisational level: general practices that provide higher quality clinical care (measured through higher quality outcomes framework (QOF) performance) are also those in which reported patient experience is better. Understanding patients' experiences of NHS Health Checks is, therefore, central to understanding the implementation of the programme, its potential impact over the first 8 years, and ways in which it might be improved to increase adherence to lifestyle advice and preventive treatments and, ultimately, improve health outcomes.

Since the introduction of the NHS Health Check programme, a growing number of both quantitative and qualitative studies reporting patients' experiences of NHS Health Checks have been published. This article provides the first systematic synthesis of these studies.

## METHODS

We performed a systematic literature review following a study protocol (available on request) that followed the Preferred Reporting Items for Systematic Reviews and Meta-Analyses (PRISMA) guidelines.

### Search strategy

We used the results of an existing literature review conducted by Public Health England (PHE) covering the period from 1 January 1996 to 9 November 2016 supplemented by a search of the Web of Science, Science Citation Index and OpenGrey covering the same period. We also hand searched the reference lists of all included publications, searched online for additional articles published by authors of the included studies and contacted the NHS Health Checks Expert Scientific and Clinical Advisory Panel to identify studies in progress or near completion. The PHE literature review included the following sources: Medline, Embase, Health Management

Information Consortium (HMIC), Cumulative Index of Nursing and Allied Health Literature Global Health, PsycInfo, the Cochrane Library, NHS Evidence, Google Scholar, Google, Clinical Trials.gov and the ISRCTN registry. Full details of all the search strategies are shown in online supplementary appendix 1. No language restrictions were applied.

### Study selection

To be included studies had to be primary research reporting the experiences of people who had attended NHS Health Checks. Commentaries, editorials and opinion pieces were excluded.

The selection of studies was performed in a two-stage process. First, the titles and abstracts were screened to identify studies relevant to the NHS Health Check. This stage had already been completed by a senior information scientist at PHE for those identified in the literature review conducted by PHE. One reviewer (EH) followed this process for the additional citations identified from the Web of Science and OpenGrey databases.

In the second stage, two researchers (JUS and AM) reviewed the full texts of all studies identified as relevant to the NHS Health Checks to select those reporting the opinions or experiences of people who had attended NHS Health Checks. Where it was unclear whether or not these inclusion criteria were met for any given study, we discussed those studies at consensus meetings with the wider research team.

### Data extraction, quality assessment and synthesis

Data on the study design, time period, recruitment methods, participants, analysis and quantitative results were extracted independently for each study by two reviewers (JUS and EH/CMa) onto data extraction forms developed to minimise bias. The quality of all each included studies was assessed at the same time using the Critical Appraisal Skills Programmes (CASP)[11] checklist for qualitative research or a checklist combining the CASP checklists for cohort studies and randomised controlled trials for the quantitative studies. We chose these checklists as they are included within the Cochrane Supplemental Guidance for Inclusion of Qualitative Research in Cochrane Systematic Reviews of Interventions[12] and identified as one of the 14 'best' tools for evaluating non-randomised quantitative studies in a review,[13] respectively, and we have successfully used them in previous reviews.[14 15] For studies that included both quantitative and qualitative methods, quality assessment was completed separately for both aspects of the study. No studies were excluded on the basis of quality alone.

As a result of the variation in methods used and experiences reported, we were unable to perform meta-analysis for the quantitative data and so synthesised that data descriptively. We synthesised the qualitative data using thematic synthesis.[16] Following reading and re-reading of the included studies, this synthesis included three stages: (1) coding of the findings of the primary studies; (2)

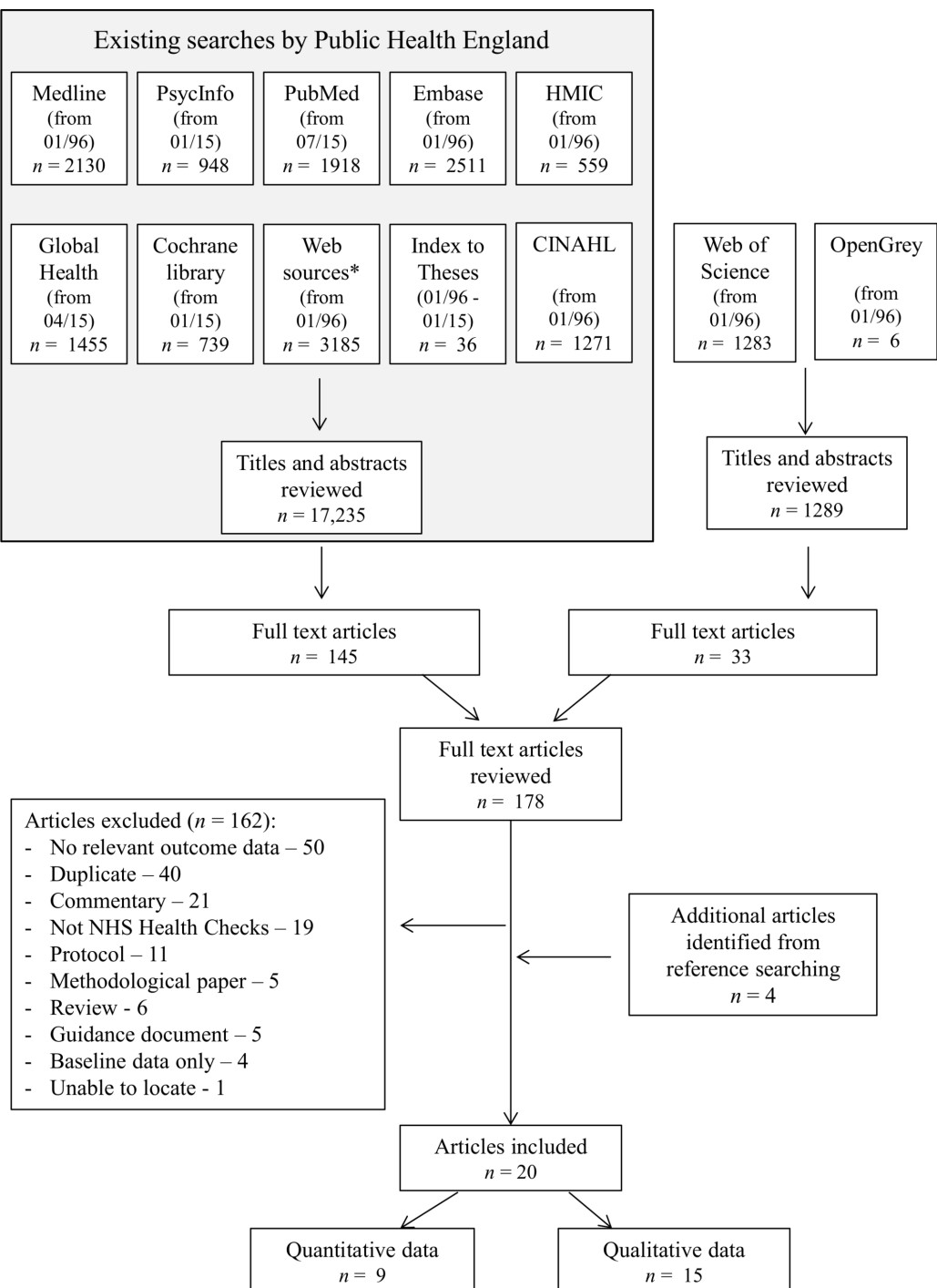

**Figure 1** PRISMA flow diagram. CINAHL, Cumulative Index of Nursing and Allied Health Literature; HMIC, Health Management Information Consortium; NHS, National Health Service; PRISMA, Preferred Reporting Items for Systematic Reviews and Meta-Analyses.

organisation of these codes into related areas to develop descriptive themes and (3) the development of analytical themes. As described by Thomas and Harden,[16] we considered all the text under the headings 'Results' or 'Findings' within the included studies as findings of the primary studies. The initial line-by-line coding of those findings was performed by at least two researchers (JUS and EH/CMa), each from a different disciplinary background (academic general practice, public services and

health systems and innovation). All have experience conducting and analysing qualitative research but none had been involved in any of the included studies. The codes resulting from that process were then discussed with members of the wider research team and the subsequent stages were an iterative process with both the descriptive and analytical themes developed through a series of meetings involving researchers from a range of clinical and non-clinical backgrounds (academic general

**Table 1** Features of studies reporting results of participant surveys

| Study/year | Type of article | Setting | n | Recruitment | Response rate (%) | Overall quality |
|---|---|---|---|---|---|---|
| Baker et al[18] | Journal article | 83 general practices | 1011 | Survey sent to all patients who had completed an NHS Health Check within a 2 month period | 43 | High |
| Corlett[25] | Journal article | Pharmacy-based NHS Health Checks | 66 | Survey sent to all those who had attended an NHS Health Check within a 4 week period | 35 | High |
| Cowper[19] | Case study | NHS Health Checks in County Durham | 483 | No details provided | Not given | Low |
| Krska et al[17] | Journal article | 16 general practices in North West England | 434 | All patients with estimated 10 year CVD risk>20% from the 16 practices were sent a postal survey regardless of whether they had attended an NHS Health Check or not | 23.4 | High |
| LGA—East Riding[22] | Case study | Outreach NHS Health Check clinics at leisure centres, community centres and workplace settings | Not given | No details provided | Not given | Low |
| NHS Greenwich[21] | Evaluation report | Outreach clinics | 540 | Questionnaire distributed at community NHS Health Check venues | Not given | Medium |
| NHS Greenwich[21] | Evaluation report | Outreach clinics | 72 | Questionnaire distributed at community NHS Health Check venues | Not given | Medium |
| 'A picture of Health'[23] | Case study | General practice-based pilot of point-of-care NHS Health Checks in Tyne and Wear | 281 | No details provided | Not given | Low |
| Taylor et al[24] | Journal article | Pharmacy-based NHS Health Checks | 97 | Pharmacists gave invitation packs to all those who attended an NHS Health Check during the first 6 months | 37.4 | High |
| Trivedy et al[20] | Journal article | Outreach NHS Health Check clinics at cricket grounds | 513 | Participants were asked to complete an anonymous questionnaire immediately after their NHS Health Check | Not given | Medium |

CVD, cardiovascular disease; LGA, Local Government Association; NHS, National Health Service.

practice, public health, health economics, clinical statistics, evidence synthesis and qualitative research). To allow an appreciation of the primary data, we have included illustrative quotations from the original studies alongside the analytical themes in this report.

A summary of the findings reported in this manuscript have been published online by Public Health England (available at http://www.healthcheck.nhs.uk/commissioners_and_providers/evidence/) and RAND (http://www.rand.org/content/dam/rand/pubs/external_publications/EP60000/EP67129/RAND_EP67129.pdf). Permission from both has been obtained to publish the results in this journal.

## RESULTS
From an initial 18 524 titles and abstracts, 178 articles were identified as potentially relevant to the NHS Health Checks and were reviewed at full-text level (figure 1). Of those, we excluded 162. The most common reasons for excluding papers were that they did not include any relevant data, were duplicates or commentaries or did not describe NHS Health Checks. Four additional articles were identified through citation searching. This review is, therefore, based on 20 articles.

### Quantitative results from patient satisfaction questionnaires
Of those 20 articles, nine include quantitative results from surveys of participants who had attended NHS Health Checks.[17–25] The details of these nine articles are shown in table 1 and full details of the quality assessment in online supplementary appendix 2 . Four are high-quality journal articles published in peer-reviewed journals in which questionnaires were sent to all those who had attended an NHS Health Check in either general practices[17 18] or pharmacies.[24 25] Response rates

were between 23.4% and 43%. A fifth study of the views of those attending outreach clinic at cricket groups was also published in a peer-reviewed journal but does not report the methods in detail or the response rate.[20] Another is a report of a service evaluation in which the views of ethnic minority participants were particularly sought[21] and the final three are low-quality case study reports which have not reported methods or response rate.[19 22 23]

The findings from those nine studies are summarised in table 2. Eight included questions about the overall experience and satisfaction with attending an NHS Health Check. Over 80% of respondents rated the experience highly or reported high levels of satisfaction. Between 86% and 99% also felt they had benefited from the NHS Health Check or would be likely or very likely to return if invited back and over 78% would recommend attendance to others. When reported (n=4), 88% to 99% of respondents felt they were given enough time. However, between 7% and 15% still had unanswered questions after the NHS Health Check.

## Qualitative data on patient experience

Patient experience was also reported in 15 qualitative studies. Three performed content analysis on

| Table 2 | Findings from studies reporting results of participant surveys |
|---|---|
| **Domain** | **Result** |
| Overall experience/ satisfaction | 91.7% rated the overall experience highly[18]<br>Almost all viewed their experience positively[25]<br>92% rated experience as good or very good[22]<br>Almost all reported a positive experience[24]<br>83% rated their experience as excellent[20]<br>82.2% were very satisfied[19]<br>97% satisfied or very satisfied overall[21]<br>95% satisfied or very satisfied[21]<br>'High levels' of satisfaction[23] |
| Recommend to others | 99.6% would recommend to others[19]<br>78% likely to recommend to others[23]<br>100% would recommend to others[20] |
| Benefit | 99% felt they had benefited[24]<br>85.6% felt they had benefited[17]<br>90.2% felt the NHS Health Check was worth attending[18]<br>90% likely or very likely to return if invited back[21] |
| Time/opportunity to ask questions | 88.0% agreed they had the time to ask questions[18]<br>92% felt they were given enough time[25]<br>94% were able to ask all their questions[25]<br>99.7% felt they were given enough time[24]<br>89.6% felt they were given enough time[17]<br>90.2% were able to ask all their questions[17]<br>9% had unanswered questions[25]<br>10.8% had unanswered questions[24]<br>14.7% still had questions about their risk of heart disease[17]<br>7.4% had concerns that had not been dealt with[17] |
| Understanding and recall of CVD risk | 97% understood everything[25]<br>59% could recall their actual CVD score[25]<br>91.9% understood everything discussed[17]<br>83% felt the health check had helped them to understand their risk of heart disease[17]<br>61.9% rated their understanding of the CVD risk score highly[18] |
| Location and timing of appointments | 69.5% rated the location of doctor's surgery highly[18]<br>70.7% rated the time and availability of appointments highly[18]<br>93.8% agreed that screening had been done in a suitable place[24]<br>86% felt the location gave enough privacy[20] |
| Staff | 93.8% rated confidence in staff knowledge[18]<br>92% reported that staff were helpful, friendly and clear about the service during their health check[22]<br>100% felt they were treated with dignity[20]<br>99% felt comfortable discussing their lifestyle[24]<br>93.6% felt comfortable discussing their lifestyle[17]<br>13.5% would have liked more support changing lifestyle[17] |

CVD, cardiovascular disease; NHS, National Health Service.

free-text responses provided in surveys[17 18 21] while the others conducted focus groups or interviews with between 8 and 45 participants. Ten are journal articles published in peer-reviewed journals,[17 18 25–32] four are research reports of service evaluations[21 33–35] and one is a master's thesis.[36] All recruited people who had attended NHS Health Checks either through invitations sent out from general practices or from community settings. Most included approximately equal numbers of men and women. Three studies had particularly sought to describe the experiences of those from ethnic minority groups.[21 31 32] In the quality assessment, 10 were high quality and 5 medium quality, with all addressing a clearly focused issue and using an appropriate qualitative method and design. The reflexivity showed the greatest variation across the studies with only five scoring medium or high for consideration of the relationship between the research team and participants. Most analysed data using thematic analysis. Further details of the design and methods used in those studies are given in table 3 and the full quality assessment in online supplementary appendix 3.

Thematic synthesis of these 15 studies identified five main analytical themes: (1) the NHS Health Check as a potential trigger for behaviour change; (2) unmet expectations; (3) limited understanding of the risk score; (4) preference for better information and (5) confusion around follow-up. The primary articles contributing to each of those themes are shown in table 4 and details of each of the themes given below.

### NHS Health Check as a potential trigger for behaviour change

Participants variously described the NHS Health Check as a 'wake-up call',[21 29] a 'reality check',[29] a 'kind of a turning point'[36] or an 'eye-opener',[32] which helped bring patients' health into focus by highlighting potential underlying health issues of which they were not necessarily cognizant[18] and making them aware that there were lifestyle-related diseases to which they may be susceptible and which they may be able to prevent.[32 34]

It's really good. It makes you aware of what problems are around. What you can get and that. It is really good. It teaches you.it's an eye-opener for people who would want to do things properly.[32]

For some, this reputedly led on to behaviour change, with many of the studies citing examples of participants who had reported making changes that they attributed to having attended the NHS Health Check.[26 29–33 37] These included changes to diet, cutting down on smoking, decreasing alcohol intake and increasing physical activity.

I've changed my diet um and, and lost a stone in weight I think as a result actually. So I'm quite happy with that, that makes me feel even healthier.[30]

Having the results of the check, I've actually started to go to (swimming baths) a couple of times, so I've made some progress….and I've actually felt better in meself.[29]

In general, dietary changes were perceived to be the easiest changes to make, particularly small changes that did not cause too much disruption to their daily routines[37] and there was recognition that changing behaviour was hard, with a number of barriers identified (box 1).

### Unmet expectations

Despite this potential influence on behaviour, a strong theme throughout many of the studies was that of unmet expectations that some participants were left with at the end of the NHS Health Check.

For many, this arose from confusion about the purpose of the NHS Health Check. The comparison made between the NHS Health Check and an 'MOT' in the promotional material and the use of the term 'Health Check' left many expecting the NHS Health Check to include a more general wide-ranging assessment of health and not just risk of cardiovascular disease.[18 26 34]

I just assumed that they would test you for everything when you were there. My perception of reading through things was that it was going to be a good overhaul, you know overall body check for everything.[34]

As a general health check it was not a series of tests as I expected. Only centred around the result of a blood test. Not comprehensive as I would have expected.[17]

Additional specific areas that participants had been expecting or thought should be covered included: a well woman check[34]; diabetes checks for all[31 34 35]; cancer screening[26 34 36]; an assessment of mental well-being[26]; an ECG[26]; testing for anaemia[26]; discussion around health conditions that impacted on their daily lives, such as joint and back pain[36] and chronic long-term conditions.[35]

### Limited understanding of the risk score

While some participants reported improved understanding of CVD risk following an NHS Health Check,[17 25 32] a common theme throughout the studies was participants' limited understanding of the risk score.

Across many of the studies there was evidence that a large number of participants were either not able to recall being provided with a risk estimate at all,[17 26 27] found the risk score confusing[18 30 34] or had interpreted it incorrectly.[26 29 30 36]

My cholesterol is high…and, I had a score saying sixteen per cent diabetes in ten years. What does that mean? I've got no idea what that means. It sounds bad because it's higher than it's meant to be but is it?[30]

My QRisk score is 11 per cent. But after getting someone to Google it for me, we still have no idea what it means. It should be explained better in a letter from the Doctor.[18]

The score itself also appeared to have little meaning or significance for most participants. Low scores (<20% 10-year estimated risk) were sometimes perceived as

**Table 3** Features of qualitative studies describing patient experiences of NHS Health Checks

| Author, year | Type of report | Location of study | Setting of NHS Health Check | Data collection method | Setting for data collection | n | Method of recruitment to study | Participant characteristics | Method of analysis | Overall quality |
|---|---|---|---|---|---|---|---|---|---|---|
| Alford[33] | Evaluation report | Knowsley | Community | Interviews and focus groups | Not given | 36 | No details given | 19 females, 17 males 13 high risk score, 23 low risk score | Thematic analysis | Medium |
| Baker et al[18] | Journal article | Gloucester | 83 general practices | Content analysis of cross-sectional survey | NA | 1011 (43%) | Survey sent to all patients who had completed an NHS Health Check within a 2 month period | 55.2% female 19% 56–60 years 10.8% 40–45 years 96% white British | Thematic analysis | High |
| Chipchase et al[34] | Report | East and North Birmingham | 2 general practices | Face-to-face semi-structured interviews | GP surgery | 10 | Attendees to NHS Health Checks in the first 2 weeks of February 2011 received a recruitment letter | 8 females, 2 males | Interpretative phenomeno-logical analysis | High |
| Corlett[25] | Journal article | London | 4 pharmacies | Telephone interviews with sample of survey respondents | On the telephone | 19 | Invitation for a semi-structured telephone interview included with survey sent to all those who had attended an NHS Health Check within a 4 week period | Not given | Thematic analysis using framework approach | Medium |
| NHS Greenwich[21] | Report | Greenwich | Community | Open-ended questionnaire, focus groups and in-depth phone interviews | Not given | 612 survey responses 4 focus groups and 31 interviews | Recruited from community outreach services providing NHS Health Checks | Ethnic minority participants: 42% female | Based on Health Belief Model | Medium |
| Ismail and Atkin[26] | Journal article | Not specified | General practices | Semistructured interviews | Participants' homes or NHS premises | 45 baseline 38 follow-up | Purposive sampling from a list provided by five participating general practices | 21 females, 24 males Average age: 58. Ethnicity: 37 White, 5 South Asian and 3 African Caribbean | Framework analysis | High |
| Jenkinson et al[27] | Journal article | Torbay | 4 general practices | Telephone or face-to-face interviews | On the telephone or participants' homes | 17 | Letters of invitation sent to a random sample identified by general practices from lists stratified by age and gender of those who had not responded to an invitation to an NHS health check within 4 weeks. | 12 females, 5 males 6 employed, 1 unemployed, 10 retired | Thematic analysis | High |
| Krska et al[17] | Journal article | Sefton, an area of North West England | 16 general practices | Postal survey with free text responses | NA | 434 (23.4%) | All patients with estimated 10 year CVD risk>20% from the 16 practices were sent a postal survey regardless of whether they had attended an NHS Health Check or not | 19% female 68.2% over 65 99.5% white 7.7% highest quintile of deprivation 13.7% lowest quintile | Categorisation of responses | Medium |
| McNaughton[37] | Journal article | North East England (non-specific location) | 5 general practices | Semistructured interviews | Not given | 29 | Invitations to patients from five general practices who had received an NHS Health Check and had an estimated 10 year CVD risk>20% | 10 females, 19 males 24 over 65 years 13 in least deprived quintile | Thematic analysis | High |
| Oswald et al[35] | Evaluation report | Teesside | General practices or pharmacies | Telephone semistructured interviews | On the telephone | 8 | Invited by general practices or pharmacies or from a list of patients who had attended an NHS Health Check and agreed to take part in the service evaluation | 6 had attended general practices and two pharmacies | Thematic analysis | Medium |

Continued

**Table 3** Continued

| Author, year | Type of report | Location of study | Setting of NHS Health Check | Data collection method | Setting for data collection | n | Method of recruitment to study | Participant characteristics | Method of analysis | Overall quality |
|---|---|---|---|---|---|---|---|---|---|---|
| Perry et al[29] | Journal article | Knowsley | Community | Interviews and focus groups | Not given | 36 | Letter or telephone invitation to all 38 people who were at high risk of CVD and had attended an NHS Health Check in the past 12–18 months were invited. The remaining attendees at low risk of CVD were purposively sampled for gender, age, risk score. | 3 focus groups: 1 for high risk scores (6 males), 2 for low risk scores (17 females and 7 males) 6 semi-structured interviews (2 females and 4 males with high risk score) | Thematic analysis | High |
| Riley et al[31] | Journal article | Bristol inner city | Community | Semistructured interviews | Community venues or participants' homes | 16 | Participants were recruited via their attendance of community outreach events. | 7 females, 9 males All from black and minority ethnic populations | Thematic analysis | High |
| Riley et al[30] | Journal article | Bristol | General practices | Face-to-face and telephone semi-structured interviews | On the telephone or in participants' homes | 28 | Purposive sampling from those identified through a search of patient records for patients who had undertaken an NHS Health Check within the previous 6 months | 16 females, 12 males 23 White British 11 most deprived quintile 11 high (>20%) CVD risk | Thematic analysis | High |
| Shaw et al[32] | Journal article | Birmingham and Black Country | General practices and community | Semistructured interviews | Not given | 23 | Patients who had attended an NHS Health Check were invited by practice managers or lead clinicians | High black and minority ethnic population and high levels of deprivation | Thematic analysis | High |
| Strutt[36] | Masters thesis | Darlington, Co Durham, UK | two general practices | Semistructured face-to-face interviews | Participants' homes | 16 | Invitation letters or telephone | 7 females, 9 males White, South-Asian and Middle Eastern | Thematic analysis | High |

CVD, cardiovascular disease; GP, general practitioner; NA, not applicable; NHS, National Health Service.

**Box 1 Reasons provided by participants for not making lifestyle changes**

► Older participants feeling that making changes to their lifestyle was unnecessary.[37]
► Healthy eating information was too generic.[37]
► Guidance they had been given was likely to be subject to change.[37]
► Comorbidities which made physical activity difficult.[37]
► Psychosocial circumstances, for example, bereavement, stress or socioeconomic barriers, such as shift work or unemployment.[29 30]
► Having previously been offered a behaviour intervention strategy.[26]
► Cost of eating fresh fruit and vegetables.[26]
► Difficulty incorporating changes into their daily lives.[29]
► Underlying medical conditions.[29]

meaning there was nothing to worry about,[29] but participants with low-risk scores were as likely to report being worried or anxious after receiving the scores as those with high-risk scores.[30] When describing their motivation to change behaviour, in general, participants also described how it was not necessarily related to their risk score and how even a high risk score was not necessarily enough to motivate them to try and change.[29 36]

> Sometimes you need a reason and I think it was like me, I needed a reason (to change) and isn't it sad that showing me the percentage wasn't reason enough for me to give up (smoking).[29]

### Preferences for better information

Most participants reported receiving lifestyle advice within the NHS Health Check. Many, however, felt it was too simple, brief, superficial or generic and felt that they would have benefited from more detail and more personalised information.[21 25 29 30 32 36 37]

> And it was that kind of information which was the kind of the bit beyond, you know, eat less, exercise more, don't smoke, don't drink. . that would have been useful. .the kind of advice that was on offer was actually very, um, simple.[30]

For some this lack of personalised information led to confusion and uncertainty,[36] with some feeling that they had received mixed messages about their health[26] and been left unsure about what actions they should take.[33] This was not a universal view, however, with some seeing the value in being provided with 'common knowledge' again as it afforded a fresh way of looking at their lifestyle and, in one study the simplicity of the information appeared to encourage participants to make changes to their behaviour.[29]

> So I thought it was very helpful it was very informative and it was thought-provoking, it just gave us some fresh view on things, because you can get very easily into doing what you think is okay.[36]

In most cases the lifestyle advice had been provided face-to-face but participants also valued, or felt it would have been helpful to have received, written information both for their own reference and also as a means to encourage behaviour change among friends and family.[18 29 34]

> Well I suppose it's good to have a question and answer thing cos you can have somebody explain it to you. But I suppose you could, something written'd be quite useful.[26]

**Table 4** Studies contributing to each of the qualitative themes

| | Unmet expectations | Limited understanding of the risk score | Preference for better information | Potential trigger for behaviour change | Confusion around follow-up |
|---|---|---|---|---|---|
| Alford[33] | | | ● | ● | ● |
| Baker *et al*[18] | ● | ● | ● | ● | |
| Chipchase *et al*[34] | ● | ● | ● | ● | |
| Corlett[25] | | ● | ● | ● | |
| NHS Greenwich[21] | | | ● | ● | ● |
| Ismail and Atkin[26] | ● | ● | | ● | ● |
| Jenkinson *et al*[27] | | ● | | | |
| Krska *et al*[17] | ● | ● | | | |
| McNaughton[37] | | | ● | ● | |
| Oswald *et al*[35] | ● | | | | ● |
| Perry *et al*[29] | | ● | ● | ● | |
| Riley *et al*[31] | | ● | ● | ● | |
| Riley *et al*[30] | ● | | | | |
| Shaw *et al*[32] | | ● | ● | ● | ● |
| Strutt[36] | ● | ● | ● | ● | |

NHS, National Health Service.

### Confusion around follow-up

The final theme related to confusion over follow-up.[21 32 33] This was particularly seen among participants who had attended NHS Health Checks in community settings. Individuals felt unsure about what steps should be taken next, specifically in relation to whether they needed to contact their general practitioner (GP) or if their GP would contact them if any causes for concern had been identified.[33] Participants also reported a lack of sufficient information on follow-up and sign posting to other NHS services.[21]

> She never said go and see your doctor. She ran out of some leaflets and she circled them and said go on the Internet.[21]

Some participants also reported that they would have liked their healthcare professionals to be more proactive in supporting them to make lifestyle changes and felt there should have been ongoing follow-up and monitoring.[26 32 35]

## DISCUSSION

### Principal findings

This study is the first systematic review of patient experience of NHS Health Checks. It shows that, among those who respond to patient satisfaction surveys, there are consistently very high levels of satisfaction with NHS Health Checks reported, with over 80% feeling that they had benefited from an NHS Health Check. However, despite these overall high levels of satisfaction, there was evidence from interviews that some participants were left with a feeling of unmet expectations. For some, this appeared to arise from confusion about the purpose of the NHS Health Check while others had been expecting a more general assessment of health. The cardiovascular risk score also appeared to generate confusion: it was poorly understood, interpreted differently among individuals with the same level of risk and seemed to have little meaning or significance for people in terms of how to use it to think about their health and future planning. Most participants reported receiving lifestyle information within the NHS Health Check but for many it was regarded as too simple and not sufficiently personalised. Nevertheless, there was evidence that the NHS Health Check was perceived to act as a wake-up call for many participants who had gone on to make lifestyle changes which they attributed to the NHS Health Check.

### Strengths and limitations

The main strengths of this review are the systematic search of multiple electronic databases, the manual searching of the reference lists of all included studies and the thematic approach to synthesis of the qualitative data. By including searches of the grey literature and the internet alongside electronic repositories, we were able to include studies that have not been published in peer-reviewed scientific journals, reducing the risk of selective reporting bias.

While we cannot exclude the possibility that additional local evaluations may have been performed or that we overlooked studies at the screening stage, we think it unlikely that they would substantially alter the main findings. Choosing to conduct a thematic synthesis for the qualitative research also ensured that we used a systematic approach to identify common themes across the studies and interpret those findings. Although some argue against the synthesis of qualitative research on the grounds that the findings of individual studies are decontextualised and the concepts identified in one setting are not applicable to another,[38] the systematic approach to coding and subsequent development of overarching themes guided by our research question enabled us to provide a synthesis of the evidence to inform practice and develop additional interpretations and conceptual insights beyond the findings of the primary studies.

The main limitations relate to the included studies. The quantitative surveys were of varying quality with response rates only reported in four of the nine studies and comparison between responders and non-responders only in one. Where reported, the response rates varied between 23% and 43%. Although survey response rates alone are a poor indicator of bias,[39 40] the included studies are, therefore, all at risk of responder bias and may represent the views of those more engaged with preventive healthcare or with particularly strong opinions. Many also measured patient satisfaction rather than patient experience. Unlike patient experience data which aim to avoid value judgements, patient satisfaction is a broad and often ill-defined concept that is multidimensional and influenced by a range of factors, including cultural norms, health status and prior experience of health care.[41] Reports of patient satisfaction can, therefore, vary widely between different patients in identical situations and in one study in which patients were asked a single question about how satisfied overall they were with their primary care practice, only 4.6% of the variance in their satisfaction ratings was a result of differences between practices.[42] Unlike patient experience data, improvements in patient satisfaction data are also not associated with improvements in care quality.[43]

The qualitative studies also included small, selected groups of participants whose expressed views are likely to be affected by both recall bias (systematic errors due to inaccuracy of recollections about NHS Health Checks) and social desirability bias (the tendency of interviewees to give responses that they think might be viewed favourably by the interviewer).[44] By virtue of the fact that they have chosen to take part in medical research the participants may also be more interested in their health than the general population so their views may not reflect the full range of views and experiences of those attending NHS Health Checks.

### Comparison with existing literature

The high levels of satisfaction reported are consistent with those reported for other NHS services, for example

the General Practice Patient Survey in which the median overall satisfaction score was 86.2.[10] The discrepancy between the very high levels of reported satisfaction in surveys and the more negative comments made in face-to-face interviews is also consistent with previous research in other areas of health care.[45–48] For example, studies have found that positive survey responses can mask important negative dimensions which patients subsequently express qualitatively[45–47] and that patients may respond differently to questions about services depending on how, where and when questions are asked.[48] The interpretation of 'good' absolute patient feedback scores should, therefore, not lead to complacency and the conclusion that improvements need not be considered.

The challenges of communicating risk are well known. Public understanding of risk is generally low and while reviews have shown that the way risk is presented affects risk perceptions,[49 50] even immediately after being provided with CVD risk information one in four people still have an inaccurate perception of their risk[51] and 1 in 10 change their perceived risk in the opposite direction to the feedback they receive.[52] The confusion around the risk scores seen in this study may therefore reflect a combination of how the risk is presented by healthcare professionals and how individuals interpret it within the context of the NHS Health Check. The finding that knowing the CVD risk score was not sufficient to motivate behaviour change is also consistent with guidance from the National Institute for Health and Clinical Excellence on behaviour change[53] and previous systematic reviews.[14 54]

### Implications for clinicians, policy-makers and future research

While participants were generally very supportive of the NHS Health Check programme and examples of behaviour change were reported, this study highlights a number of areas where improvements could be made. In particular, the finding that a number of patients were unaware of the programme or had misunderstood the extent and purpose of the NHS Health Check suggests that more proactive communications may be needed to raise awareness of the programme overall and that patients need additional clarity about the programme when being invited. The expectation that the NHS Health Check would be a more general health check also raises questions about whether the programme should be expanded to cover other areas of health. The lack of clarity around follow-up and reports that participants would have liked their healthcare professionals to be more proactive in supporting them to make lifestyle changes additionally suggests that there are potential missed opportunities to support behaviour change. This may be in part due to a lack of appropriate services to refer patients to but may also reflect a need for additional training for those delivering the NHS Health Check. Finally, the confusion around the CVD risk score among patients highlights the potential limitations of relying on the risk score alone as a trigger for facilitating behaviour change within NHS Health Checks and the need for adequate training and time for healthcare professionals to help patients understand their

risk in line with best practice guidance.[50 55] Further research is also needed to determine whether different communication strategies, such as heart age,[56] improve understanding and subsequent behaviour change.

**Acknowledgements** We thank our patient and public representatives Kathryn Lawrence and Chris Robertson for providing helpful comments on the findings and the NHS Health Checks Expert Scientific and Clinical Advisory Panel working group for providing us with the initial literature search conducted by Public Health England. We would also like to thank Anna Knack, Research Assistant at RAND Europe, for her excellent research support and Emma Pitchforth for her helpful comments on our analysis.

**Contributors** JUS developed the protocol, screened articles for inclusion, extracted and synthesised the quantitative and qualitative data, interpreted the findings and wrote the first draft of the manuscript. EH screened articles for inclusion, extracted and synthesised the qualitative data, interpreted the findings and critically revised the manuscript. CMa extracted and synthesised the qualitative data and critically revised the manuscript. AM screened articles for inclusion, interpreted the findings and critically revised the manuscript. CS, CM, FW, SG and JM developed the protocol, interpreted the findings and critically revised the manuscript.

**Funding** This work was funded by a grant from Public Health England. JUS was funded by an NIHR Clinical Lectureship and FW by an NIHR Clinician Scientist award. The views expressed in this publication are those of the authors and not necessarily those of the NHS, the NIHR or the Department of Health. All researchers were independent of the funding body and the funder had no role in data collection, analysis and interpretation of data, in the writing of the report or decision to submit the article for publication.

**Disclaimer** The corresponding author has the right to grant on behalf of all authors and does grant on behalf of all authors, an exclusive licence on a worldwide basis to the BMJ Publishing Group Ltd and its Licensees to permit this article (if accepted) to be published in BMJ editions and any other BMJPGL products and sublicences to exploit all subsidiary rights, as set out in our licence (). All authors had full access to all of the data in the study and can take responsibility for the integrity of the data and the accuracy of the data analysis The corresponding author affirms that the manuscript is an honest, accurate and transparent account of the study being reported; that no important aspects of the study have been omitted and that any discrepancies from the study as planned (and, if relevant, registered) have been explained.

**Competing interests** None declared.

**Provenance and peer review** Not commissioned; externally peer reviewed.

**Data sharing statement** All data are available from the reports or authors of the primary research. No additional data are available.

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
