## [Reviewer comments · BMJ Open]

ARTICLE DETAILS

TITLE (PROVISIONAL)	Patient experience of NHS Health Checks: a systematic review and qualitative synthesis
AUTHORS	Usher-Smith, Juliet; Harte, Emma; MacLure, Calum; Martin, Adam; Saunders, Catherine; Meads, Catherine; Walter, Fiona; Griffin, Simon; Mant, Jonathan

VERSION 1 - REVIEW

REVIEWER	Ruth Riley University of Bristol UK
REVIEW RETURNED	11-Apr-2017

GENERAL COMMENTS	Reviewer's Comments Reflexivity: Include further information about the background/discipline of the coders and their relationship with data/papers under review Information sources: No contact with study authors to identify additional studies – perhaps you could justify why this was not undertaken Assessment criteria: In Table 2 and in your assessment criteria, I would include information about where the interviews/focus groups were carried out e.g. in the patients' home as this may influence their responses. Similarly, there is no inclusion of the theoretical framework employed in the qualitative studies under review or reference to such in the assessment criteria. Could you include a summary paragraph regarding the quality of included papers in reference to the CASP tool i.e. a narrative summary of Appendix 3 – this would provide the reader with reassurances regarding your methods of assessment and overview of the quality of included studies
---

REVIEWER	Rebekah McNaughton Teesside University, England
REVIEW RETURNED	20-Apr-2017

GENERAL COMMENTS	I feel this is an interesting review that prioritises the experiences of those who attend the NHSHC as a lot of the evidence so far relates to how to get people in for the check and not how those people are looked after during and after assessment. Below are a few comments on the manuscript, they are intended to make this
---

submission clearer and stronger.

1. line 28 (abstract): insert NHSHC before the word 'process' to be clear what process you are describing. This also relates throughout.

2. The introduction seems a little bit light on references to the NHSHC programme (policy documents etc...). Also, line 15: there is a brief sentence about 'risk tools' I feel you need to expand on how the cardiovascular risk score is calculated. It is important for later findings to acknowledge that the risk score is a synthesis of surrogate markers for CVD - it does not identify any 'illnesses' rather it identifies 'risk' - it would be good to explain how the risk tools do this.

3. lines 26 and 27: you talk about the use of finite resources and how there is discussion about if the NHSHC is a good use of these resources? There is no reference to who is having these discussions? You need to reference appropriate sources.

4. lines 31-51: you discuss how patient experience impact on many aspects of care. However, experience of the NHSHC will also impact on compliance (or not) to lifestyle advice and prophylactic medications. Without compliance all the NHSHC is doing is identifying at risk people not getting those at risk to change anything!

5. Data extraction, quality assessment, and synthesis: was quality assessment not carried out before data extraction (therefore making sure you were not taking 'low quality' evidence through to synthesis?) if so, should it be reported first? line 11-12 in this section (for some reason on my document line numbers restart after 60 which is confusing!) you provide reference [7] for the qualitative CASP tool but do not reference the quantitative ones.

6. line 18: you need to provide a reference after "...thematic synthesis" to which framework for synthesis you used.

7. there is no reference to how you synthesised the quantitative evidence?

8. I am unclear about how you synthesised the qualitative evidence. I would refer you to the work of Mary Dixon-Woods about qualitative systematic reviewing - did you extract the themes that were derived from the authors for your own synthesis OR did you extract the 'raw' data from the papers and perform your own thematic analysis. what are the strengths and limitations of whichever approach you used?

9. Results: you state on line 13 that high levels of satisfaction are reported? How is satisfaction defined? is it satisfaction with
- the 'service received'?
- the NHSHC offering?
- the way the assessment was conducted?
- the outcome of assessment?
- did all papers report the same definition of 'satisfaction'?

10. qualitative synthesis. section '1': are the quotation terms for the check the author's or participant's words here? it is unclear - and pretty important.

11. same section, line 58: you state [the check brings]...health into focus by highlighting underlying health issues of which they were not

	congnizant... I think a more accurate statement would be that the check highlighted POTENTIAL underlying health issues - this is a risk assessment, it does not provide diagnosis. 12. section 5 'confusion around follow-up': there is no reference to the annual review process here - feels like a big part of the NHSHC offering that is being ignored 13. the last comment under the 'principal findings' section states that 'participants had gone on to make SUBSTANTIAL changes...' there is no evidence presented in this review that changes had been "substantial". Some stated they had made changes - but were these substantial in nature? 14. I feel that the strengths and limitations section should be presented at the end - it cuts up discussion in a strange way presented here - though this may be a journal structure requirement 15. you use technical terms such as recall bias and social desirability you should reference appropriate sources for these. 16. you state at the end of the limitations section that participants were probably more interested in their own health - they could also have been incentivised to take part in the research? 17. comparison with existing literature, line 31: please provide references to the 'previous research' you point to. 18. same section line 38: " under the direct control of the person they are evaluating". This doesn't make sense? are the participants evaluating the researchers? Or do you mean that the evaluation team have no control over the NHSHC programme therefore participants don't raise concerns? 19. Same section: lines 44-50: you discuss that relying on the risk score to be a catalyst for change is not sufficient. Is it that being told you are at risk isn't enough or is it the way in which risk is communicated by health professionals to at risk individuals is not sufficient. David Spiegelhalter has done loads of work about risk communication around CVD. Maybe it is the way risk is presented doesn't make sense to people? Or is it that people don't internalise risk as they would a diagnosis of cancer, for example? I think there are much deeper issues at play here.
--	---

VERSION 1 – AUTHOR RESPONSE

Reviewer: 1

Ruth Riley

University of Bristol, UK

Please state any competing interests or state 'None declared': None declared

Please leave your comments for the authors below Reviewer's Comments

Reflexivity: Include further information about the background/discipline of the coders and their relationship with data/papers under review

We have added the following sentences to the methods section:

“The initial coding of the findings of the primary studies was performed by at least two researchers (JUS + EH/CMa), each from a different disciplinary background (academic general practice, public services, and health systems and innovation). All have experience conducting and analysing qualitative research but none had been involved in any of the included studies. These findings were then discussed with members of the wider research team and the subsequent stages were an iterative process with both the descriptive and analytical themes developed through a series of meetings involving researchers from a range of clinical and non-clinical backgrounds (academic general practice, public health, health economics, clinical statistics, evidence synthesis and qualitative research). To allow an appreciation of the primary data, we have included illustrative quotations from the original studies alongside the analytical themes in this report.”

Information sources: No contact with study authors to identify additional studies – perhaps you could justify why this was not undertaken

While we did not contact study authors to identify additional studies, we did hand search the reference lists of all included publications and search specifically for additional studies published by the authors of the included studies. We also contacted the NHS Health Checks Expert Scientific and Clinical Advisory Panel to identify any studies in progress or near completion that they were aware of. To make this clearer in the manuscript we have added the following sentences to the ‘Search Strategy’ section of the methods:

“We used the results of an existing literature review conducted by Public Health England covering the period from 1st January 1996 to 9th November 2016 supplemented by a search of the Web of Science, Science Citation Index and OpenGrey covering the same period. We also hand searched the reference lists of all included publications, searched online for additional articles published by authors of the included studies, and contacted the NHS Health Checks Expert Scientific and Clinical Advisory Panel to identify studies in progress or near completion.”

Assessment criteria: In Table 2 and in your assessment criteria, I would include information about where the interviews/focus groups were carried out e.g. in the patients’ home as this may influence their responses. Similarly, there is no inclusion of the theoretical framework employed in the qualitative studies under review or reference to such in the assessment criteria.

As suggested by the reviewer we have added two columns to Table 2 including information about where the interviews were carried out and what analysis method was used.

Could you include a summary paragraph regarding the quality of included papers in reference to the CASP tool i.e. a narrative summary of Appendix 3 – this would provide the reader with reassurances regarding your methods of assessment and overview of the quality of included studies

We agree with the reviewer that a summary paragraph describing the quality of the included qualitative papers would be helpful. We have accordingly added the following section to the results:

“Patient experience was also reported in 15 qualitative studies. Three performed content analysis on free-text responses provided in surveys^{17,18,21} whilst the others conducted focus groups or interviews with between 8 and 45 participants. Ten are journal articles published in peer reviewed journals^{17,18,25–32} four are research reports of service evaluations^{21,33–35}, and one is a Masters thesis³⁶. All recruited people who had attended NHS Health Checks either through invitations sent out from general practices or from community settings. Most included approximately equal numbers of men and women. Three studies had particularly sought to describe the experiences of those from ethnic minority groups^{21,31,32}. In the quality assessment, ten were high quality and five medium

quality, with all addressing a clearly focused issue and using an appropriate qualitative method and design. The reflexivity showed the greatest variation across the studies with only five scoring medium or high for consideration of the relationship between the research team and participants. Most analysed data using thematic analysis. Further details of the design and methods used in those studies are given in Table 2 and the full quality assessment in Appendix 3.”

Reviewer: 2

Rebekah McNaughton

Teesside University, England

Please state any competing interests or state 'None declared': None declared.

Please leave your comments for the authors below I feel this is an interesting review that prioritises the experiences of those who attend the NHSHC as a lot of the evidence so far relates to how to get people in for the check and not how those people are looked after during and after assessment.

We are pleased the reviewer feels our manuscript is an interesting review covering an area which has been less well reported.

Below are a few comments on the manuscript, they are intended to make this submission clearer and stronger.

1. line 28 (abstract): insert NHSHC before the word 'process' to be clear what process you are describing. This also relates throughout.

We thank the reviewer for this suggestion and to be completely clear have replaced the word 'process' with 'an NHS Health Check' in this line and also throughout the manuscript.

2. The introduction seems a little bit light on references to the NHSHC programme (policy documents etc...). Also, line 15: there is a brief sentence about 'risk tools' I feel you need to expand on how the cardiovascular risk score is calculated. It is important for later findings to acknowledge that the risk score is a synthesis of surrogate markers for CVD - it does not identify any 'illnesses' rather it identifies 'risk' - it would be good to explain how the risk tools do this.

We have added reference to the Best Practice Guidance for the NHS Health Check Programme. We also agree that providing more background about the risk scores would be helpful so have added the following text to the first paragraph of the introduction:

“The NHS Health Check itself consists of three components: risk assessment, communication of risk and risk management¹. For CVD the QRISK[®]2 risk tool² is first used to estimate the individual’s risk of developing CVD based on risk factors including age, sex, ethnicity, smoking status, height and weight, family history of coronary heart disease, blood pressure, and cholesterol. That estimated risk, expressed as the percentage risk of developing disease over the next 10 years, is then used to raise awareness of relevant risk factors and inform discussion about the lifestyle and medical approaches best suited to managing the individual’s risk of disease. Risk assessment for diabetes was introduced in 2016 and patients at high risk of developing type 2 diabetes who should receive a screening blood test are identified either by using either validated risk assessment tools or a diabetes filter¹.”

3. lines 26 and 27: you talk about the use of finite resources and how there is discussion about if the NHSHC is a good use of these resources? There is no reference to who is having these discussions? You need to reference appropriate sources.

We are grateful to the reviewer for highlighting the need for references here. We have added reference to the following two articles:

Capewell S, McCartney M, Holland W. Invited debate: NHS Health Checks-a naked emperor? *J Public Health (Oxf)* 2015;37:187–92.

Dalton AR, Marshall T, McManus RJ. The NHS Health Check programme: a comparison against established standards for screening. *Br J Gen Pract* 2014;64:530–1.

4. lines 31-51: you discuss how patient experience impact on many aspects of care. However, experience of the NHSHC will also impact on compliance (or not) to lifestyle advice and prophylactic medications. Without compliance all the NHSHC is doing is identifying at risk people not getting those at risk to change anything!

We are grateful to the reviewer for this additional reason why patient experience of the NHS Health Checks is important. We have added reference to it as below:

“Understanding patients’ experiences of NHS Health Checks is, therefore, central to understanding the implementation of the programme, its potential impact over the first eight years, and ways in which it might be improved to increase adherence to lifestyle advice and preventive treatments and, ultimately, improve health outcomes.”

5. Data extraction, quality assessment , and synthesis: was quality assessment not carried out before data extraction (therefore making sure you were not taking 'low quality' evidence through to synthesis?) if so, should it be reported first? line11-12 in this section (for some reason on my document line numbers restart after 60 which is confusing!) you provide reference [7] for the qualitative CASP tool but do not reference the quants ones.

In order to provide a comprehensive overview of the research in this area, we did not exclude any studies based on quality alone and performed quality assessment at the same time as the data extraction. The reference [7] for the qualitative CASP tool is also the same reference as for the quantitative tools. To clarify these points we have amended the section to read:

“The quality of all included studies was assessed at the same time using the Critical Appraisal Skills Programmes (CASP)⁷ checklist for qualitative research⁷ or a checklist combining the CASP checklists for cohort studies and randomised-controlled trials for the quantitative studies. For studies that included both quantitative and qualitative methods, quality assessment was completed separately for both aspects of the study. No studies were excluded on the basis of quality alone.”

6. line 18: you need to provide a reference after "...thematic synthesis" to which framework for synthesis you used.

We performed thematic synthesis of the qualitative studies as described by Thomas and Harden. We had included that reference in the following sentence but have moved it to immediately follow the term 'thematic synthesis' for clarity.

7. there is no reference to how you synthesised the quantitative evidence?

We performed a descriptive synthesis of the quantitative evidence. The variation in methods used across the studies and different experiences reported meant that meta-analysis was not possible. To clarify this we have added the following sentence to the methods section:

“As a result of the variation in methods use and experiences reported, we were unable to perform meta-analysis for the quantitative data and so synthesised that data descriptively.”

8. I am unclear about how you synthesised the qualitative evidence. I would refer you to the work of Mary Dixon-Woods about qualitative systematic reviewing - did you extract the themes that were derived from the authors for your own synthesis OR did you extract the 'raw' data from the papers and perform your own thematic analysis. what are the strengths and limitations of whichever approach you used?

As mentioned above under comment 6 we synthesised the qualitative studies using thematic synthesis as described by Thomas and Harden (BMC Medical Research Methodology 2008). As described in that paper, we considered all the text under the heading 'Results' or 'Findings' in the studies as primary data and coded that line by line. We then organised those codes into first descriptive themes and then combined those descriptive themes to derive overarching analytical themes. To make this clearer in the manuscript we have amended the methods section to read:

“We synthesised the qualitative data using thematic synthesis¹⁶. Following reading and re-reading of the included studies, this synthesis included three stages: 1) coding of the findings of the primary studies; 2) organisation of these codes into related areas to develop descriptive themes; and 3) the development of analytical themes. As described by Thomas and Harden¹⁶, we considered all the text under the headings 'Results' or 'Findings' within the included studies as findings of the primary studies. The initial line-by-line coding of those findings was performed by at least two researchers (JUS + EH/CMa), each from a different disciplinary background (academic general practice, public services, and health systems and innovation). All have experience conducting and analysing qualitative research but none had been involved in any of the included studies. The codes resulting from that process were then discussed with members of the wider research team and the subsequent stages were an iterative process with both the descriptive and analytical themes developed through a series of meetings involving researchers from a range of clinical and non-clinical backgrounds (academic general practice, public health, health economics, clinical statistics, evidence synthesis and qualitative research). To allow an appreciation of the primary data, we have included illustrative quotations from the original studies alongside the analytical themes in this report.”

The strengths of that approach are included in the following text in the strengths and limitations section of the discussion:

“Choosing to conduct a thematic synthesis for the qualitative research also ensured that we used a systematic approach to identify common themes across the studies and interpret those findings. Although some argue against the synthesis of qualitative research on the grounds that the findings of individual studies are de-contextualised and the concepts identified in one setting are not applicable to another³⁸, the systematic approach to coding and subsequent development of overarching themes guided by our research question enabled us not only to provide a synthesis of the evidence to inform practice but also to develop additional interpretations and conceptual insights beyond the findings of the primary studies.”

9. Results: you state on line 13 that high levels of satisfaction are reported? How is satisfaction defined? is it satisfaction with

- the 'service received'?
- the NHSHC offering?
- the way the assessment was conducted?
- the outcome of assessment?
- did all papers report the same definition of 'satisfaction'?

We agree with the reviewer that our previous description of the findings from the studies reporting participant surveys was brief and was potentially unclear. In this revised version we have removed the findings from those studies from Table 1 and moved them into a new Table 2. In that new Table 2 we have presented the findings under each domain reported within the surveys rather than for each study separately. This allows the reader to easily see the findings across the studies. As in the earlier version we have also used the same wording as in each of the primary studies for each of the findings so that the Table reflects the different ways the questions were asked to participants.

We have also edited the description of the quantitative findings in the results section so that it now reads:

“The findings from those nine studies are summarised in Table 2. Eight included questions about the overall experience and satisfaction with attending an NHS Health Check. Over 80% of respondents rated the experience highly or reported high levels of satisfaction. Between 86% and 99% also felt they had benefited from the NHS Health Check or would be likely or very likely to return if invited back and over 78% would recommend attendance to others. When reported (n=4), 88% to 99% of respondents felt they were given enough time. However, between 7% and 15% still had unanswered questions after the NHS Health Check.”

10. qualitative synthesis. section '1': are the quotation terms for the check the author's or participant's words here? it is unclear - and pretty important.

We thank the reviewer for highlighting the importance of distinguishing between the authors' interpretations and the participants' words within the studies. We have amended the start of this section to only include in quotations phrases that were used by participants. All text in quotations and italics within the manuscript is therefore from participants.

“Participants variously described the NHS Health Check as a “wake-up call”^{17,25}, a “reality check”²⁵, a “kind of a turning point”³², or an “eye-opener”²⁸, which helped bring patients' health....”

11. same section, line 58: you state [the check brings]...health into focus by highlighting underlying health issues of which they were not cognizant... I think a more accurate statement would be that the check highlighted POTENTIAL underlying health issues - this is a risk assessment, it does not provide diagnosis.

We agree with the reviewer and have added the word 'potential' to the text. It now reads:

“...which helped bring patients' health into focus by highlighting potential underlying health issues of which they were not necessarily cognizant¹⁰...”

12. section 5 'confusion around follow-up': there is no reference to the annual review process here - feels like a big part of the NHSHC offering that is being ignored

13. the last comment under the 'principal findings' section states that ' participants had gone on to make SUBSTANTIAL changes...' there is no evidence presented in this review that changes had been "substantial". Some stated they had made changes - but were these substantial in nature?

We agree with the reviewer that our choice of the term 'substantial' was inappropriate. We have deleted the term from that sentence so that it now reads:

“Most participants reported receiving lifestyle information within the NHS Health Check but for many it was regarded as too simple and not sufficiently personalised. Nevertheless, there was evidence that

the NHS Health Check was perceived to act as a wake-up call for many participants who had gone on to make lifestyle changes which they attributed to the NHS Health Check.”

14. I feel that the strengths and limitations section should be presented at the end - it cuts up discussion in a strange way presented here - though this may be a journal structure requirement

The journal guidelines suggest the discussion follows the overall structure: a statement of the principal findings; strengths and weaknesses of the study; strengths and weaknesses in relation to other studies, discussing important differences in results; the meaning of the study: possible explanations and implications for clinicians and policymakers; and unanswered questions and future research. We have therefore left the strengths and limitations section in the same place but would be happy to move this if felt to be more appropriate by the editorial team.

15. you use technical terms such as recall bias and social desirability you should reference appropriate sources for these.

We have added the following reference which describes both recall bias and social desirability bias:

Althubaiti A. Information bias in health research: definition, pitfalls, and adjustment methods. *J Multidiscip Healthc* 2016;9:211–7.

16. you state at the end of the limitations section that participants were probably more interested in their own health - they could also have been incentivised to take part in the research?

On re-reading of all the included studies there is no mention in any of them about participant incentives for taking part. We therefore have no evidence on which to comment on whether this was a factor or not.

17. comparison with existing literature, line 31: please provide references to the 'previous research' you point to.

The 'previous research' referred to in that sentence is that which is fully described, including references, in the following sentences. We have added those references again after 'previous research' and are happy for the editorial team to decide where they would best be cited. Those sentences now read:

“The discrepancy between the very high levels of reported satisfaction in surveys and the more negative comments made in face-to-face interviews is also consistent with previous research in other areas of health care^{37–40}. For example, studies have found that positive survey responses can mask important negative dimensions which patients subsequently express qualitatively^{37–39} and that patients may respond differently to questions about services depending on how, where and when questions are asked⁴⁰.”

18. same section line 38: " under the direct control of the person they are evaluating". This doesn't make sense? are the participants evaluating the researchers? Or do you mean that the evaluation team have no control over the NHSHC programme therefore participants don't raise concerns?

The phrase “that patients often give positive satisfaction ratings even in the context of a negative experience when they believe the poor care is not under the direct control of the person they are evaluating” refers to situations in which patients do not feel that the poor care they have received or experienced is under the control of the person who they have been asked to evaluate. We agree with the reviewer that, as the participants in the studies included in this review were asked to evaluate their

experience and satisfaction with the NHS Health Check programme and not individual providers, this is less likely to be relevant in this context. To remove any confusion we have therefore deleted that phrase. The sentence now reads:

“For example, studies have found that positive survey responses can mask important negative dimensions which patients subsequently express qualitatively^{37–39} and that patients may respond differently to questions about services depending on how, where and when questions are asked⁴⁰.”

19. Same section: lines 44-50: you discuss that relying on the risk score to be a catalyst for change is not sufficient. Is it that being told you are at risk isn't enough or is it the way in which risk is communicated by health professionals to at risk individuals is not sufficient. David Spiegelhalter has done loads of work about risk communication around CVD. Maybe it is the way risk is presented doesn't make sense to people? Or is it that people don't internalise risk as they would a diagnosis of cancer, for example? I think there are much deeper issues at play here.

We completely agree with the reviewer that there are likely deeper issues at play here and that the confusion reported around the risk scores is likely a combination of both how that risk is communicated by health professionals and also how individuals appraise and process that risk information. We have amended lines 44-50 to reflect some of those issues:

“The challenges of communicating risk are well known. Public understanding of risk is generally low and whilst reviews have shown that the way risk is presented affects risk perceptions^{45,46}, even immediately after being provided with CVD risk information one in four people still have an inaccurate perception of their risk⁴⁷ and one in ten change their perceived risk in the opposite direction to the feedback they receive⁴⁸. The confusion around the risk scores seen in this study may therefore reflect a combination of how the risk is presented by healthcare professionals and how individuals interpret it within the context of the NHS Health Check. The finding that knowing the CVD risk score was not sufficient to motivate behaviour change is also consistent with guidance from the National Institute for Health and Clinical Excellence (NICE) on behaviour change⁴⁹ and previous systematic reviews^{50,51}.”

VERSION 2 – REVIEW

REVIEWER	Ruth Riley University of Bristol UK
REVIEW RETURNED	17-May-2017

GENERAL COMMENTS	I have read the author's revisions and responses and am satisfied that they have adequately addressed my comments.
--

REVIEWER	Rebekah McNaughton Teesside University, England.
REVIEW RETURNED	09-May-2017

GENERAL COMMENTS	All of my comments were attended to.
--------------------------------------